# Graph-based algorithms for Laplace transformed coalescence time distributions

**Gertjan Bisschop**  *

University of Edinburgh, Institute of Evolution and Ecology, Edinburgh, United Kingdom

* g.bisschop@sms.ed.ac.uk

## Abstract

Extracting information on the selective and demographic past of populations that is contained in samples of genome sequences requires a description of the distribution of the underlying genealogies. Using the Laplace transform, this distribution can be generated with a simple recursive procedure, regardless of model complexity. Assuming an infinite-sites mutation model, the probability of observing specific configurations of linked variants within small haplotype blocks can be recovered from the Laplace transform of the joint distribution of branch lengths. However, the repeated differentiation required to compute these probabilities has proven to be a serious computational bottleneck in earlier implementations.

Here, I show that the state space diagram can be turned into a computational graph, allowing efficient evaluation of the Laplace transform by means of a graph traversal algorithm. This general algorithm can, for example, be applied to tabulate the likelihoods of mutational configurations in non-recombining blocks. This work provides a crucial speed up for existing composite likelihood approaches that rely on the joint distribution of branch lengths to fit isolation with migration models and estimate the parameters of selective sweeps. The associated software is available as an open-source Python library, `agemo`.

**Data Availability Statement:** The software is available at http://github.com/LohseLab/agemo. Documentation on how to use agemo can be found at https://agemo.readthedocs.io.

## Author summary

For simple models of idealised populations, the process that generates the observed sequences can be mathematically described. For a given number of samples, we can enumerate all possible genealogies. We can even incorporate the impact of past events like population size reductions on the observed sequence variation. However, the number of possible genealogies will become very large, very fast. So, to extract information from the observed mutations, we need mathematical tools and efficient algorithms to use the information contained within the large collection of possible genealogies.

The Laplace transform is one such mathematical tool that allows us to recursively generate the branch length distribution of all genealogies. Here I show how the transform can be represented as a graph. Using this nonlinear data structure, I define a general procedure to efficiently evaluate the associated mathematical expressions. And I further show how this can be used to speed up existing composite likelihood approaches to fit

**Funding:** This work was funded by an ERC starting grant awarded to Konrad Lohse ((ModelGenomLand, 757648, erc.europa.eu)). The funders had no role in study design, data collection and analysis, decision to publish, or preparation of the manuscript.

**Competing interests:** The authors have declared that no competing interests exist.

demographic models and estimate sweep parameters. The associated software, `agemo`, has a well-documented Python API and has been designed with extensibility in mind, making it an ideal back-end for many other inference approaches in population genetics.

This is a *PLOS Computational Biology* Software paper.

## Introduction

The Laplace transform is often introduced as a formal tool to solve differential equations. Yet in mathematical fields like queuing theory, the integral transform is used to simplify the analysis of the studied probabilistic problems. This is due to two key properties of the Laplace. One, the transform of the distribution of the sum of independent random random variables becomes the product of their respective Laplace transformed marginal distributions. Two, the Laplace transform of a random variable $X$ describing the length of an interval, has a clear probabilistic interpretation. Let a Poisson process with intensity $\omega$ mark this interval, then the Laplace transform, $\mathcal{L}(f_X)(\omega) = f_X^\star = E[e^{-\omega X}]$, is the probability of not observing any marks in the considered interval [1]. If we translate this idea to the standard coalescent framework [2–4] and let the marking process describe the arrival of mutations, then the Laplace transform of the distribution of the total branch length gives the probability of not observing any mutations along the modelled genealogies. Conversely, the probability of not observing any mutations between two coalescence events gives us the transformed distribution of the total branch length spanned by those two events. This means that, given a state-space graph that describes all possible transitions during the coalescent process, one can easily write down all associated expressions in the Laplace domain [5]. This is not the case for the time domain.

The Laplace transformed description of the distribution of coalescence times has been used to tackle two major problems in population genetics: fitting explicit models of population history [5–8] as well as estimating sweep parameters [9]. Note that because the Laplace transform can be interpreted as the generating function (GF) of a continuous random variable, this method is often referred to as the GF approach. These studies have used composite likelihood-based approaches that summarize mutational information as counts of the (joint) site frequency spectrum within blocks of a fixed length [7]. Note that the GF framework allows for the inclusion of multiple recombining loci [5]. However, because of the associated computational complexity, current applications all calculate likelihoods based on the mutational information in blocks small enough for recombination to be negligible. Also note that likelihoods are not only composited over all blocks, but also over all possible subsamples of size at most 6 [8]. This is due to the fact that the GF grows superexponentially with the number of samples, but also because of the number of possible joint site frequency spectra within small blocks (see Mutation configuration probabilities).

The probabilities of observing all block-wise mutational configurations are given by a multivariate Poisson distribution mixed over the joint distribution of branch lengths. The Laplace transform is well suited to compute these probabilities because this Poisson distribution can be written as a function of the Laplace transform of the branch length distribution. Specifically, the probability of observing $\mathbf{k}$ mutations along each of the $k_i$ branch types is proportional to the $\mathbf{k}^{th}$ derivative with respect to the associated variable in the Laplace domain [5]. Previous implementations have always used a computer algebra system (CAS) to compute these higher-

order derivatives of the Laplace transform. Such an approach suffers from an explosion in the number of terms due to the product or the Leibniz rule. This computational bottleneck has limited the usability of the framework.

One way to solve the computational bottleneck is to replace the recursive description of the generating function by matrix manipulations. This has been done using phase-type theory [10]. Phase-type distributions are the result of a mixture or convolution of exponential distributions. The theory provides an alternate way of translating the state transition diagram into a description of the branch length distribution of genealogical trees. One of the major computational advantages of this description is that the matrices that define the distribution are typically preserved up to the point of evaluation [10]. Because computationally efficient algorithms for linear algebra operations already exist, phase-type theory lends itself to efficient implementations. With increasing sample size however, matrix operations become computationally unfeasible. Given the sparsity of most real-world state spaces, graph-based representations of these matrices can alleviate this issue to some extent [11]. However, currently no algorithms have been described to extract information from the joint branch length distribution.

Graphs reveal the relationships between the base entities or nodes. Therefore, the more complex these relationships are, the more connected the graph will be and the more useful a graph-based representation becomes. In machine learning for example, computational graphs representing mathematical equations are used to efficiently calculate derivatives [12]. What I have implemented here takes inspiration from automatic differentiation in that we will use the recursively generated state-space graph as a computational graph to avoid both symbolic computation and repetitively evaluating the same expressions. Note that representing the GF as a graph does not remove the exponential growth with sample size of both the state space and the number of possible mutational configurations within small blocks.

Here, I present the key algorithms underlying this approach as well as `agemo`, an open-source non-symbolic implementation of the GF framework. The paper is structured as follows. First, I will summarize the description of the GF as a large symbolic expression as defined in [5] and show how the GF can be represented more succinctly. Secondly, I lay out the graph traversal algorithm that allows efficient evaluation of the GF. I then show how this algorithm can be used to query the joint distribution of branch lengths and tabulate the probabilities of observing mutation configuration in blocks of non-recombining sequence.

## Methods

### Recursive description of the branch length distribution in the Laplace domain

Given a sample of $n = \sum_{i=1}^{k} n_i$ uniquely labeled lineages from $k$ distinct populations, we can represent all possible coalescent histories of that sample in a single rooted directed graph [13]. By labelling lineages by the samples they subtend, we can associate each node of the transition diagram with a vector $\Omega = (\Omega_1, \ldots, \Omega_k)$ uniquely describing that state. Here $\Omega_i$ represents all lineages present in deme $i$. As we move through the graph from the source node, representing the set of all sampled lineages, to the absorbing state or most recent common ancestor (mrca), we track the movement and coalescence of lineages. Coalescence events reduce the number of lineages in a deme, while events like a (mass) migration will move lineages from one deme to another. Fig 1 shows the state space graph for a toy example. Here, edges 1 and 3 are associated with mass migration events moving all lineages from population $B$ to $A$, leaving $\Omega_B$ empty. Note that, in general, any state where all but $\Omega_i$ are empty and $|\Omega_i| = 1$, can be an absorbing state.

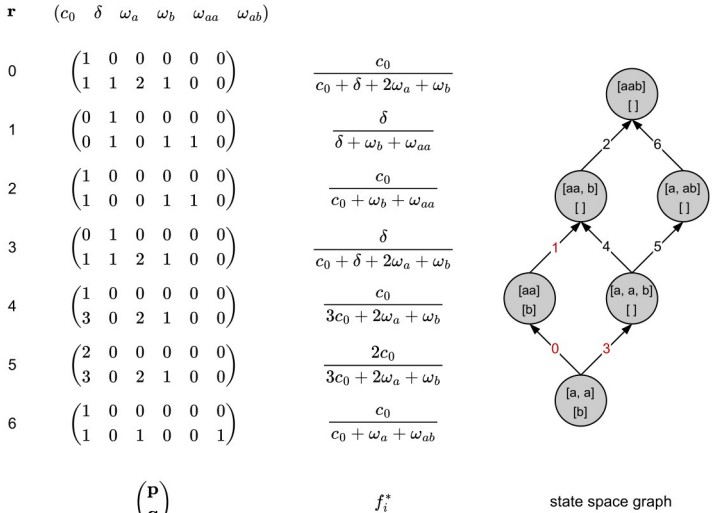

**Fig 1. Coalescent state space graph for two populations $A$ and $B$ with 2 and 1 unphased sample(s), respectively.**
Each node in the graph is labeled by the lineages present in each population as indicated by the square brackets ($\Omega_A$: top, $\Omega_B$: bottom). The demographic model assumes a single mass migration event from population $B$ to $A$ back in time. Vector $r$ consists of the rates of all possible events: coalescence ($c_0$) in population $A$, a dummy variable $\delta$ associated with the mass migration (see Discrete events), and a dummy variable $\omega_k$ for each possible branch type. The Laplace transform associated with edge $i$, $f_i^\star$, can be retrieved as the ratio of the dotproduct of $p$ and $q$ with $r$. Red indices on the graph indicate edges associated with an equation containing $\delta$. Edges 1 and 3 represent a mass migration event, moving all lineages from $B$ to $A$. All other transitions represent coalescence events. All paths can be described by enumerating the indices of the associated matrices: $\mathcal{P} = ((0, 1, 2), (3, 4, 2), (3, 5, 6))$.

The state space graph as described above can be generated recursively. All state transitions are conditionally independent. They only depend on the lineages in the current state and on the set of competing processes that define how one moves from one state to the next. Along each path through the graph, the time to the mrca is distributed as the sum of the inter-event times. In the Laplace domain, the sum of independent random variables is equal to the product of their respective Laplace transformed distributions. This general property of generating functions turns this graph into more than just a visual aid. The graph is now a description of how to generate the Laplace transform $f_L^\star$, of the random variable $L$, describing all branch lengths. Given the expressions that detail the time to go from one state to the next, the distribution associated with a single path through the graph can be retrieved by multiplying all the expressions associated with the edges found along that path. The entire Laplace transform is then a simple sum of the equations describing all paths through the graph. A minimal representation of the GF thus consists of a list of the unique equations, each associated with a single edge, and a list of lists $\mathcal{P}$ enumerating for each path $b$ through the graph the index $a$ of each edge/equation along that path (see Fig 1). As a result $f_L^\star$ can be written as $\sum_b \prod_{a \in \mathcal{P}_b} f_a^\star$.

Because of the probabilistic interpretation of the Laplace transform, the expression associated with each edge equates to the probability of observing the event of interest before any other event happening at rate $\omega$. In the standard coalescent framework, coalescence events are exponentially distributed with rate $\binom{n}{2}$ when there are $n$ lineages remaining. So in the Laplace domain, the distribution of the waiting time until the next coalescence is given by $\binom{n}{2}/(\binom{n}{2} + \omega)$. To incorporate more than one process with an exponentially distributed waiting time, it suffices to observe that $min(X, Y) \sim exp(\omega_x + \omega_y)$ when both $X \sim exp(\omega_x)$ and $Y \sim exp(\omega_y)$. This means one can incorporate as many events with exponentially distributed

waiting times as computationally possible. The probability that the event with rate $\lambda_i$ happens before the other $j - 1$ competing events will still have the same general form.

$$f^\star(\boldsymbol{\lambda}, \boldsymbol{\omega}) = \frac{l_i\lambda_i}{\sum_j l_j\lambda_j + \sum_k o_k\omega_k} \qquad \text{with } 0 < i \le j \tag{1}$$

In this equation, we associate a unique dummy variable ($\omega_k$) with each of the $k$ branch types along which all $j$ competing processes ($\lambda_j$) happen. Roman letters represent integers that count the number of branches of a particular type ($o_k$), or the number of ways a certain (coalescence or other) event can modify the current state ($l_j$). Note that in the case of multiple populations, coalescence rates are given relative to the rate in a reference population, i.e. $\lambda_i = N_{e_{ref}}/N_{e_i}$.

Let $\boldsymbol{r} = (\boldsymbol{\lambda}, \boldsymbol{\omega})' = (\lambda_1, \ldots, \lambda_i, \ldots, \lambda_j, \omega_1, \ldots, \omega_k)'$, denote the vector containing the rates of all $j$ competing processes as well as the $k$ dummy variables, and let $\boldsymbol{p} = (0, \ldots, l_i, \ldots, 0)$ and $\boldsymbol{q} = (l_1, \ldots, l_i, \ldots, l_j, o_1, \ldots, o_k)$, then Eq 1 can be written as the ratio of $\boldsymbol{p} * \boldsymbol{r}$ and $\boldsymbol{q} * \boldsymbol{r}$. Looking at our example (Fig 1), edge 0 is associated with a coalescence event of two lineages in $A$. There is only 1 possible way these lineages can coalesce, so the first entry of $p_0$ is 1. $q_0$ encodes all possible events (coalescence or mass migration) given the set of sampled lineages and a count of each of the branch types present prior to coalescence. Starting out, we have two $a$ lineages and a single $b$ lineage. All other equations, and their corresponding arrays can be deduced in the same way.

Storing the equations in this way ensures that we can efficiently substitute in parameter values by taking the dot product with a vector $\boldsymbol{r}$ representing a point in parameter space once the Laplace transform needs to be evaluated. Also, storing the equation coefficients in matrix form allows us to efficiently perform operations on the equations (see Discrete events).

**Discrete events.** For the general description of the GF, we have assumed that all competing processes have exponentially distributed waiting times. As outlined by [5], discrete events that only happen once can be included by treating them initially as a competing exponentially distributed process with rate $\delta$. The Laplace transform of the joint branch length distribution $g^\star_{L|T}$, given the discrete event happened at time $T$, can be recovered by taking the inverse transform of the GF as described in the previous section, $f^\star_L(\boldsymbol{\omega}, \delta)$, divided by its associated dummy variable, $\delta$. To see this, note that $f^\star_L$ can be written as a compound distribution integrating over the probability density function of the time to the discrete event: $f^\star_L(\boldsymbol{\omega}, \delta) = \int_0^\infty \delta e^{-\delta T} g^\star_{L|T}(\boldsymbol{\omega}, T)dT$. Dividing both sides by $\delta$ we recognize $\frac{1}{\delta}f^\star_L(\boldsymbol{\omega}, \delta) = \mathcal{L}(g^\star_{L|T})(\delta)$. This procedure has been used to incorporate population divergence, admixture events, and bottlenecks [5, 7], as well as selective sweeps [9].

Previous implementations have relied on a CAS to obtain an analytic solution for the inverse transform of the GF. However, as long as we limit ourselves to a single discrete event, the GF will always be a sum of the products of factors of the form $f^*(c_i) = \frac{1}{c_i+\delta}$. Using partial fraction expansion, we can formulate a closed-form solution, to the inverse Laplace, $\mathcal{L}^{-1}(f^*)$, with respect to $\delta$, where $T$ represents the time to the discrete event. Also, having stored all equation coefficients as an array, we can do so in a way that allows for efficient substitution of all parameter values.

$$\mathcal{L}^{-1}\left(\prod_{i=0}^k f^\star(c_i)\right) = (-1)^{k+1}\sum_{i=0}^k \frac{e^{-c_iT}}{\prod_{\substack{j=0 \\ i\neq j}}^k (c_i - c_j)} \tag{2}$$

Looking at a single path along the graph, only the equations associated with edges leading up to the node representing the discrete event will contain $\delta$. Equations associated with edges

past that point can be treated as constants. The resulting inverse of this path will therefore be an expression given by Eq 2 times the unchanged equations associated with all edges positioned downstream, i.e. moving through the graph backwards in time, of the node associated with the discrete event.

**Adding in new events.** Currently, in addition to coalescence, two types of events have been implemented in the Python library `agemo`: unidirectional migration at a constant rate and population splits (forwards in time). Because of the recursive description, adding in more event types is straightforward and only requires a description of all possible state transitions due to that event given the current configuration of lineages. Note that the library does currently not accommodate events that generate cycles in the graph. This means that bi-directional migration, for example, is not supported. This does not mean that the GF framework cannot handle events that generate cycles. Using a Taylor series expansion, the GF can be decomposed into histories with $1, 2, \ldots, m$ cyclic events [8].

## Graph traversal algorithm

The one-to-one correspondence between the state space graph and the Laplace transform means the state space graph can be thought of as a computational graph. Evaluating the transform at a single point $s$ in the Laplace domain equates to substituting the value into the expression associated with each edge, followed by multiplying the results along each path and adding the results across all paths. This is the general idea that will be used in the next paragraph.

However, note that because of the general form of the inverse in the case of a discrete event (Eq 2), the graph needs to be modified slightly so that, ultimately, the nodes of the computational graph again represent the factors of a multiplication. This is achieved by, for each path, reducing the part of the path leading up to the node associated with the discrete event to a single edge and pairing that edge with the result of Eq 2. This is demonstrated in Fig 2A and 2B. Edges marked with red indices are associated with equations containing the dummy variable setting the rate of the discrete event and require inversion. Edge 0 and 1 are collapsed into a single edge and are now associated with $\mathcal{L}^{-1}(f_0^\star * f_1^\star)$ as defined in Eq 2. Also note that to simplify the evaluation algorithm, equations are represented by nodes instead of edges (Fig 2C).

**General algorithm.** Given the computational graph (as described in Graph traversal algorithm and see Fig 2C), a general algorithm to propagate any evaluation of the equations associated with each node is given by Alg 1. The algorithm relies on the fact that, implicitly, the edges of the graph represent multiplication. The evaluated values associated with each node do not need to be single floats. They can be the coefficients of a generating function, for example, representing the probabilities of seeing particular mutation types (see Mutation configuration probabilities). In these cases, multiplication and addition operators will need to be defined for propagation. We can then rely on the commutative property to efficiently traverse the graph towards the root. Especially in the case where addition is a less costly operation than multiplication (as is the case for polynomials, see Mutation configuration probabilities), it will pay off to add the values associated with the children of a node prior to multiplication. The annotations on Fig 2C demonstrate the algorithm for our toy example.

**Algorithm 1**: Propagate values through graph

```
1 function PROPAGATE;
  Input: A (adjacency list of graph), N (evaluated equations for each
         node), E (topological sorting of graph)
2 foreach parent in E do
3   children = A[parent];
4   temp = 0;
5   if children then
```

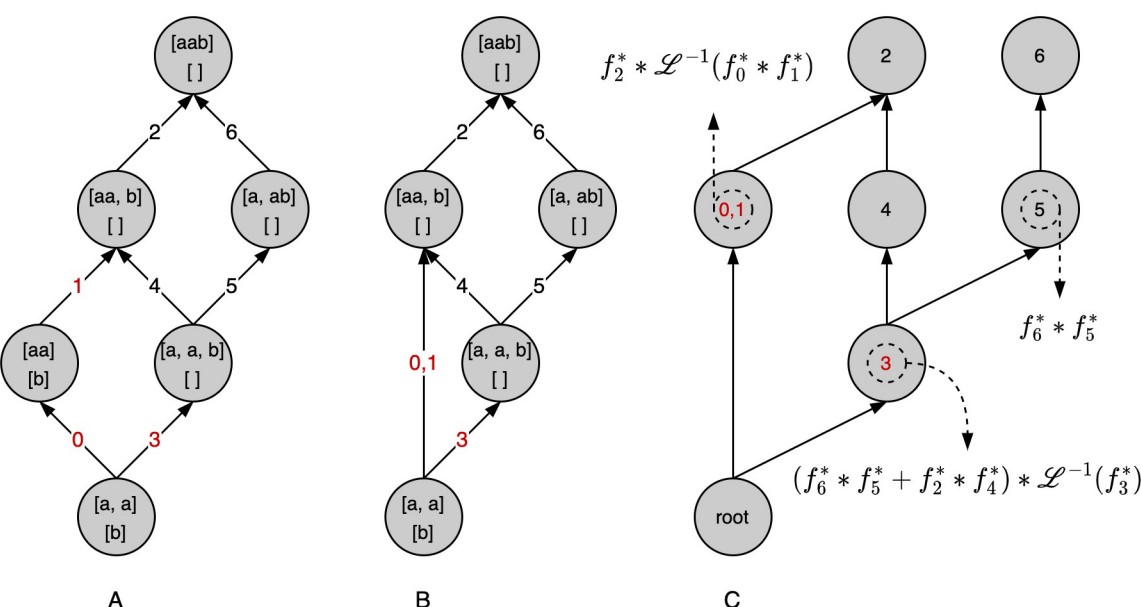

**Fig 2. From coalescent state space to computational graph: State space graph and model identical to Fig 1.** A: Unmodified state-space graph. B: Collapsed form, grouping all parts of each path that require inverting with respect to the dummy variable associated with the discrete event. The integers in red are the indices of the equations containing $\delta$. C: To simplify the formulated algorithms, nodes represent the equations previously associated with the edges. The graph has been annotated to demonstrate the general propagation algorithm of the evaluated equations associated with each node towards the root. $\mathcal{L}^{-1}(f_i^\star)$ is the inverse Laplace of $f_i^\star$ with respect to $\delta$.

```
6    foreach child in children do
7       temp+=N[child];
8    end
9    if parent not root then
10      N[parent] = PRODUCT(temp, N[parent]);
11    else
12      return temp;
13    end
14   end
15 end
```

**Mutation configuration probabilities.** Assuming branches are labelled by the samples they subtend, $2^n - 2$ branch types can be distinguished for a sample of $n$ lineages. Along each of these branch types, mutations might occur. Under an infinite-sites mutation model, the joint probability of seeing $k_i$ mutations along each of these $i$ branch types in short blocks of a given length can be derived using the GF. Each mutation configuration is then defined as a vector of the form $(k_1, \ldots, k_{2^n-2})$ where each entry is a count within the interval $[0, 1, 2, \ldots, k_i^{max} + 1]$. $k_i^{max} + 1$ is used to group all mutation configurations with more than $k_i^{max}$ mutations. The array representing all possible block-wise mutation configuration counts is of size $\prod_i (k_i^{max} + 2)$.

This is a more general description of the block-wise site frequency spectrum or bSFS as introduced by [7]. The bSFS only distinguishes mutations along branches with the same number of descendants. By ignoring both phase and root information, all mutation configurations are essentially instances of the folded (joint) site frequency spectrum for blocks of a fixed length. The bSFS is a tally of all observed (joint) SFS instances. Note that the absence of phase

and root information can be accommodated via a simple relabeling of all branches, and therefore their associated dummy variables in the GF [8]. By labeling all samples by the population they were collected from, one can incorporate unknown phase. Removing root information is identical to the concept of folding the SFS, i.e. combining branch types on either side of the root.

Quite often, the array containing the probabilities associated with all block-wise mutation configurations will be sparse. Some branch types can simply never be jointly observed along any of the possible genealogies. As a consequence, we know the probability of a configuration indicating the presence of mutations along these branch types will always be zero without having to perform any computations. agemo therefore pre-determines incompatible branch types and only reserves memory and performs computations for mutation configurations with a non-zero probability. This implies both time and significant memory savings.

In the presence of phase information further savings can be made. Because of the symmetries inherent to the coalescent, some mutation configurations can be equally likely. For example, in a single population with samples $(a, b, c)$ observing a single mutation along both branches $ab$ and $b$ is equiprobable to observing a single mutation along both branches $bc$ and $c$. Here again, agemo will only compute the probability of a single representative of the set of all equiprobable mutation configurations.

The probability of observing mutation configuration **k** under a specified model is proportional to a term in a (truncated) Taylor series expansion (see Eq (1) in [5] for details). Any naive approach, based on calculating all higher order derivatives using a CAS, will suffer from an explosion in the number of terms due to the Leibniz or product rule when differentiating. Generally, a CAS will fail to take into account the fact that the same partial derivatives of the functions that constitute the expression are computed multiple times. This problem has been well studied for the purpose of automatic differentiation algorithms [14–17]. In fact, it has been shown that a set of recurrence relations can be defined on the coefficients of truncated Taylor series to efficiently compute higher-order derivatives [18]. Departing from the elementary functions as represented by a computational graph, a complex Taylor series expansion can be performed without recalculating the same derivatives.

**Algorithm 2**: Product of two truncated Taylor series [18].

```
1 function SERIES PRODUCT;
  Input: Two arrays A and B with same shape
  Output: array C of same shape as A and B
2 foreach multi-index k < shape(A) do
3   sum = 0;
4   foreach multi-index j ≤ multi-index k do
5     sum = ADD(sum, A[j] * B[k − j]);
6   end
7   C[k] = sum;
8 end
9 return C;
```

Translating this to the graph traversal algorithm outlined above requires us to first obtain all coefficients for a truncated Taylor series of the equation associated with each node in our computational graph. We can then use the algorithm defining the product of two truncated Taylor series (see Alg 8 in [18] and Alg 2) to propagate the coefficients of the series associated with each node. Note that adding two truncated Taylor series simply amounts to the pairwise addition of all corresponding coefficients. To obtain the higher-order derivatives needed for the first step, we could use the recurrence relations defined in [18]. Note however, that the computational graph representation of the GF we have constructed is not at the level of the elementary functions. Because all equations associated with each of the nodes are well

characterized, we can define a closed-form implementation of the derivatives with respect to the distinguished branch types. The equations all fall into one of two categories, depending on whether an inversion step was needed. Given a first-degree multivariate polynomial of the form $f(\mathbf{x}) = \sum_i c_i x_i + b$, non-inverted equations can be written as $1/f(\mathbf{x})$. Inverted equations on the other hand have building blocks that take on the form of $e^{cf(\mathbf{x})}/(\prod_j f_j(\mathbf{x}))$. Using Alg 2 and Eq 4, we can come up with all partial derivatives for the inverted equations as well. With $s = \sum_i k_i$ and $\mathbf{x}$ representing the branch type vector,

$$\frac{\partial f(\mathbf{x})^{-1}}{\partial^{k_i} x_i} = (-1)^s s! \frac{c_i^{k_i}}{f(\mathbf{x})^2} \tag{3}$$

$$\frac{\partial e^{cf(\mathbf{x})}}{\partial^{k_i} x_i} = c^s c_i^{k_i} e^{cf(\mathbf{x})} \tag{4}$$

Note that Alg 2 contains an explicit ADD function. Care needs to be taken when summing (a subset of) the coefficients of a Taylor series: these will be both positive and negative, and as such, catastrophic cancellation might occur, leading to accuracy loss. To counteract this, I implemented the compensated summation algorithm of Ogita-Rump-Oishi [19]. The loss of precision is bounded by keeping track of small errors and adjusting the result using the error term. An alternative way of handling this would be to temporarily increase numeric precision at the crucial steps. Lastly, an advantage of using Taylor series coefficients rather than the corresponding derivatives is that the coefficients will always be smaller by a factor $(\sum \mathbf{k})!$, leading to less cumulative rounding error [18].

## Results and discussion

The work presented here constitutes a CAS-independent, open-source implementation of the GF approach. A general outline has been given on how the correspondence between the event state-space graph and the GF can be used to query the distribution of Laplace-transformed coalescence times efficiently. In particular, an algorithm has been laid out to calculate the probability of block-wise mutation configurations by propagating the calculation of series coefficients down the graph of ancestry states. The fact that this automation does not rely on a CAS and that it has been implemented in Python makes `agemo` an ideal back-end for likelihood calculations.

`agemo` relies on `numba` [20] just-in-time compilation to speed up the critical parts of the code. Compiling the code using `numba` has a few consequences. Firstly, compilation happens the first time the code is run and the resulting compiled code is written into a file-based cache. Secondly, some numerical operations are implemented differently in `numba` than in `numpy`. In the case of summation this can lead to a loss of precision and has required the implementation of a compensated sum algorithm. A potentially faster solution would be to temporarily increase machine precision for the evaluation of particular sums. However, this is not possible using `numba` and would therefore require translating part of the code to C.

To evaluate accuracy and performance, I calculated the bSFS for an isolation with migration (IM) model with 2 populations and 2 lineages in each population. Here two populations are descended from a common ancestral population at some time in the past, and since then unidirectional gene flow is assumed to have happened at a constant rate [21]. When discarding root and phase information, this leaves just 4 branch types. For each branch type I set $k_{max} = 2$, which means that the final result will contain $4^4$ elements. This is the most complex model for which there exists a CAS-based implementation. Note that the original Mathematica implementation [5] can only calculate the bSFS for a simplified IM model with two $N_e$ parameters,

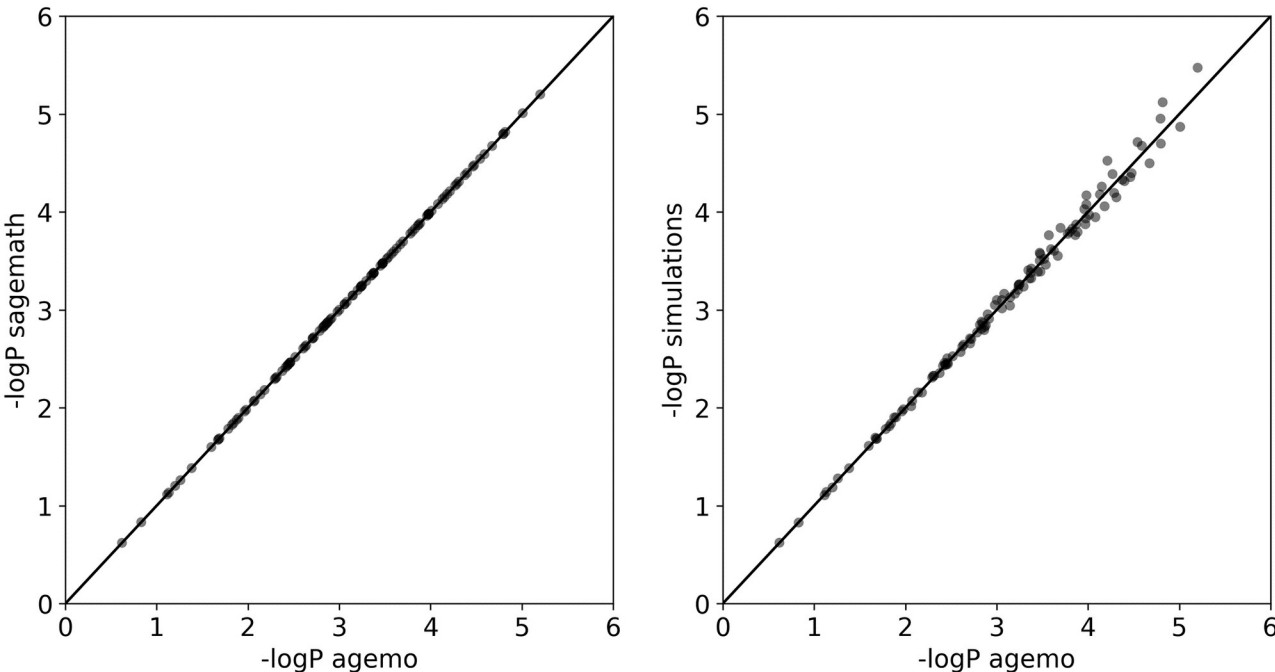

**Fig 3. Accuracy: These scatter plots compare the negative logarithm of the computed probability ($-logP$) of observing each block-wise mutation configuration with a non-zero probability.** The output of an independent `Sagemath` implementation (left) or of the Monte Carlo simulation-based approach [23] (right) is plotted against the output of `agemo`. 1000 simulation replicates were used. Model parameters: IM model, 2 samples per population, migration from A to B (backwards in time). $N_{e_{AB}} = 1.5e6$, $N_{e_A} = 1.3e6$, $N_{e_B} = 0.6e6$, $m_e = 7e-7$, $T = 1e7$, $\theta = 1.152$, $k_{max} = (2, 2, 2, 2)$.

meaning that at least two populations must have the same size. An implementation using open source CAS Sagemath [22] takes about 75 s for a non-simplified model where each population has a unique $N_e$, while `agemo` evaluates a single point in parameter space in 181 ms. Increasing the sample size to 3 lineages in each population increases the number of nodes in the graph from 76 to 4449. The bSFS now contains $4^7$ elements. Run time goes up accordingly to 134 s. The Python module contains multiple test suites testing all functions. Accuracy is assessed against an independent Sagemath implementation (Fig 3 left).

This IM model can be simplified to only include migration. We assume migration has been going on for an infinitely long time. Without any discrete events the graph is now maximally connected. For 2 lineages per population `agemo` takes 5 ms. Table 1 shows how performance scales with an increase in the number of samples per population.

I also benchmarked the evaluation time and accuracy (Fig 3 right) against the simulation-based approach as described in [23]. Using `msprime` [24], coalescent trees can be simulated under (almost) any demographic model. Without having to simulate mutations, we can calculate the probability of observing each mutation type. Given that mutations on each branch type happen independently, the probability of seeing mutation configuration $(k_1, k_2, \ldots, k_n)$ is

**Table 1. Run times migration-only model, 2 populations, unphased and unrooted branchtypes, $k_{max} = 2$.**

| samples per population | nodes in graph | non-zero entries in bSFS | size bSFS | time |
| --- | --- | --- | --- | --- |
| 2 | 30 | 112 | 256 | 5 ms |
| 3 | 196 | 1408 | 16384 | 480 ms |
| 4 | 1106 | 21952 | 16777216 | 52.3 s |

given by the product of $n$ probabilities as given by a Poisson distribution with rate $\theta/2 * t_i$. Here, $t_i$ is the total branch length of branch type $i$ and $\theta = 4N_e\mu$. When averaged across many replicates, the true value will be approximated. Note that particular entries of the bSFS might require fewer/more replicates to get at a good approximation than others [23, 25]. For the IM-model, with 1000 replicates one can already approach the true bSFS quite well [23] (see also Fig 3). Scaling linearly with the number of replicates, this takes about 450 ms. For 3 lineages per population run time goes up to 917 ms. With 4 lineages per population, there are $4^{12}$ entries in the bSFS, making a non-sparse approach prohibitively slow. Using simulations, run times are the same for both the IM and the migration model. Note that I aimed to make the comparison as fair as possible by optimizing the code and compiling the critical parts with `numba`. All calculations have been done on the same MacbookPro (2.2 GHz 6-Core Intel Core i7). Also note that the bottleneck of simulating the bSFS is not the actual simulation itself but the inherent combinatorial explosion of an ever increasing number of mutation configurations with increasing sample size. I attempted to alleviate this by means of sparse matrices, but this came at a speed cost.

This issue is inherent to the way the array of block-wise mutation configuration counts is defined and also applies to `agemo`. In part, this is solved by only calculating and storing the values associated with each unique mutation configuration that has a non-zero probability. However, computing the residual probabilities (observing more than $k_{max_i}$ mutations along each branch type $i$) in the last step currently still requires us to populate an array of size $\prod(k_{max_i} + 2)$. Memory usage quickly becomes an issue here, and solving this requires a general sparse array implementation of the existing function. This suggests that the mutation configuration counts array would benefit from a dedicated sparse-data structure. Ideally, this data structure would also enable us to take advantage of the dependency structure of all higher-order derivatives.

A last inherent limitation to the GF approach is that although we can include discrete events, retrieving the expression parametrized by the time to that discrete event requires us to take an inverse Laplace transform. Unfortunately, translating the mathematical description into a computational graph does not simplify this issue. As discussed, with the inclusion of discrete events the state space graph can no longer be translated into a computational graph without modification. A node must be added to the computational graph for each path leading to a discrete event, thus increasing the number of nodes and decreasing the connectivity of the graph, making a graph-based approach less efficient. `agemo` will therefore always do better in scenarios without discrete events (Table 1).

As indicated in the Methods section, extending the GF approach to include new event types can easily be done. Because of its recursive nature, it only requires defining a function that describes the impact of the event on the extant lineages. All implemented events can then be combined to define a structured coalescent model. Note however that the current implementation only contains closed-form expressions to efficiently evaluate the GF associated with at most a single discrete event.

The general algorithm outlined here should enable users to query the Laplace transform to extract, for example, topology information, the SFS or the time to the first coalescence event. These functionalities have not been explicitly implemented yet. But, they can be computed using the described graph and associated expressions. Also, `agemo` was designed with extensibility in mind. Future work on this library will enable a more diverse range of structured coalescent models as well as the ability to dynamically restrict the graph to those paths that are compatible with a specified topology.

The work described in this paper shows significant similarities with recent progress in phase-type theory [11]. The authors present a general graph-based description of multivariate phase-type distributions and demonstrate the ability of their approach by calculating the SFS for an IM-type model sampling 7 lineages from each population. There are two main advantages of the phase-type theoretic approach. Firstly, incorporating discrete events does not require taking an inverse transform. Second, the paper contains algorithms using Gaussian elimination to translate cyclic graphs into an acyclic phase-type distribution, thus taking care of issues associated with, for example, bi-directional migration. Dealing with bi-directional migration thus no longer requires a costly matrix inversion.

On the other hand, `agemo` allows users to take full advantage of the information present within the joint distribution of coalescence times (e.g. block-wise mutation configuration counts). Including short-range linkage information comes at a computational cost, limiting the applicability to smaller sample sizes. However, previous work has demonstrated that this approach maximizes the information contained in small samples compared to relying on the SFS [7, 9]. More importantly however, both frameworks have (independently) combined the same two basic ingredients to efficiently describe coalescent models: a recursive state-space construction and a graph representation for fast evaluation of the represented distributions.

## Acknowledgments

I would like to thank Konrad Lohse, Derek Setter, Kevin Thornton and Helen Alexander for constructive comments on the manuscript.

## Author Contributions

**Conceptualization:** Gertjan Bisschop.

**Investigation:** Gertjan Bisschop.

**Software:** Gertjan Bisschop.

**Writing – original draft:** Gertjan Bisschop.

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
