## [Decision Letter · Decision Letter 0]

28 Jun 2022

Dear Mr Bisschop,

Thank you very much for submitting your manuscript "Graph-based algorithms for Laplace transformed coalescence time distributions." for consideration at PLOS Computational Biology. As with all papers reviewed by the journal, your manuscript was reviewed by members of the editorial board and by several independent reviewers. The reviewers appreciated the attention to an important topic. Based on the reviews, we are likely to accept this manuscript for publication, providing that you modify the manuscript according to the review recommendations.

Two of the reviewers see great value in the new algorithm presented in the study. Reviewer 2's view is that there is too much focus on algorithms and not enough on biological questions; the reviewer also found the notation problematic and poorly defined, making the paper difficult to follow. Both Reviewer 1 and Reviewer 2 remark that the accuracy of the method should be checked in some way. We agree that this would be useful. Overall, given how widely the coalescent is used and how significant it can be to improve computational efficiency in inference methods for complex biological questions, the potential usefulness of this paper is clear. However, the flaws in the notation, figures and general presentation need to be addressed; please note suggestions made by all three reviewers.

Sincerely,

Mark M. Tanaka

Associate Editor

PLOS Computational Biology

William Noble

Deputy Editor

PLOS Computational Biology

[LINK]

Two of the reviewers see great value in the new algorithm presented in the study. Reviewer 2's view is that there is too much focus on algorithms and not enough on biological questions; the reviewer also found the notation problematic and poorly defined, making the paper difficult to follow. Both Reviewer 1 and Reviewer 2 remark that the accuracy of the method should be checked in some way. We agree that this would be useful. Overall, given how widely the coalescent is used and how significant it can be to improve computational efficiency in inference methods for complex biological questions, the potential usefulness of this paper is clear. However, the flaws in the notation, figures and general presentation need to be addressed; please note suggestions made by all three reviewers.

Reviewer's Responses to Questions

**Comments to the Authors:**

Reviewer #1: In this paper Bisschop develops a more efficient algorithm to obtain the probability of observing a particular mutational configuration at a non-recombining locus by using a graphical representation of generating function computations. The material is obviously technical and the contributions of the present work are clearly explained. This is an important and difficult problem in population genetics, and the present manuscript represents a nice advance. I have a number of minor comments below.

I prefer to sign my reviews,

Jeffrey P. Spence

Minor comments:

The author should feel free to ignore this if they consider it to be too far outside the scope of the paper, but I imagine that it would considerably improve the impact of the work to show some real-world implications of the speedup offered by the present method. As an example, the present method allows for scaling to larger sample sizes. Does that buy anything in terms of inference? Does the Monte Carlo error in the simulation-based methods used in Beeravolu et al. 2018 result in worse inference compared to exact computation? These questions could be addressed using a small simulation study and one of the previously defined inference tasks.

In retrospect it is obvious, but I was struck by the results in Table 1 for how difficult the generating function is to compute for even relatively small sample sizes (e.g., computing likelihoods for 5 diploids seems like it would be prohibitively expensive). I think it would be good to mention in the introduction the overall limitations of the generating function approach. While the present work certainly makes this approach more scalable, it would be good to state up front that it is extending something that applies to only extremely small sample sizes to something that can be applied to sample sizes that would still be considered small in most cases.

A bit of additional background would be nice around lines 115-117 regarding how to obtain the generating function in the case of one or more discrete events. The material is present in Lohse et al. (2011) but it would make the present paper easier to follow if enough of the relevant material was included to be self-contained.

It would be good to include a figure or table to show that the calculated bSFSs match those computed using previous generating function (or phase-type distribution theory), just to show that the implementation is correct and there are no issues with numerical precision.

More explanation around equation 2 is required. What is g? What is \\mathcal{L} and what is its inverse?

Both the alternating signs and the division by a difference in equation (2) suggests that the formula could be quite numerically unstable in certain regimes. It is mentioned a bit later in the manuscript, but how does agemo deal with the numerical instability? Can it detect if it's in a regime where higher numerical precision is needed?

Throughout it's assumed that all processes are time-homogeneous. This is violated in the (common) case of time-varying population sizes. Can inhomogeneous processes be modeled or must they be represented as piecewise homogeneous and then dealt with as "discrete events"? Similarly, how does the computation scale as the number of discrete events increases? Is it feasible to allow populations sizes to change a few times while also modeling a simple IM scenario?

The caption in Figure 2 refers to flipping the role of nodes and edges in a graph as "inverting". I have only heard of the "inverse" of a graph in the context of the complement of a graph -- where nodes in the inverse graph have an edge between them if and only if they do not have an edge between them in the original graph, which is quite different from the usage here.

As a very minor comment -- numba allows caching computations (e.g., adding `cache=True` to any @njit calls) which may be able to prevent the cost of repeated compilation discussed on line 219-220.

Typos:

line 30: "respect to associated variable"  "respect to the associated variable"

Reviewer #2: This article describes a new software implementation for computing the Laplace transform associated with coalescence time distributions, under demographic models that allow for a certain amount of structure, including ongoing or discrete migration events between demes. The code is available from Github.

Although the paper falls within the field of computational biology, it does not seem well suited to this journal. There is a great focus on algorithmic detail but little on any specific biological question. There is a missing 'second half' of this paper which would apply the software to an interesting biological problem and to provide further insight. This would probably require some downstream implementation which includes how to associate the Laplace transform to likelihood computations by considering placement of mutations along branches. I would also like to see more careful benchmarking - not just in runtime, but some evidence that the output is accurate. Altogether, as it stands this work might be more suited as an 'application note' at a venue such as Bioinformatics.

The description of the algorithm is very hard to follow in several places. Partly this is due to mathematical imprecision. To give one example, equation (1) seems to be fundamental to the paper, but we are not told what l_i' represents, nor why the right-hand side seems to depend on index i (but not the left), nor what is m, nor what each summand is over. The phrase 'branch type' is used here and throughout the paper without a careful definition. Equation (2) is similarly afflicted - I'm afraid I could not work out what any of g, script-L, or, f_i are supposed to represent, nor what are the ranges of all of the summands. I can make a reasonable guess for some of them, but I don't think a typical reader should have to. In aggregate the accumulation of missing details like this make the paper very difficult to follow. At the very least there should be a complete, mathematically precise description of what is going on either in the main text or in a supplement, with simple worked examples.

The accompanying figures do not help very much. Figure 1 seems to use completely different notation to the main text, and the graph in that figure is not explained at all.

I am sorry to sound so negative. I think the software could be useful to researchers in this area, and it is clear that a lot of work has gone into getting the coding right. It is just that, from this paper, one cannot see the forest for the trees.

Minor comments:

- Figure referencing has not worked properly. e.g. Fig 1 is called Fig 2.1 in some places.

- Some references are incomplete (lines 321, 328, 330, 373).

Reviewer #3: This paper describes an efficient method for computing the distribution of branch lengths, and hence, of the probabilities of sets of mutations in a non-recombining region. This will make it possible to use the blockwise site frequency spectrum (bSFS) without using symbolic computation, allowing the method to be widely used. This is a substantial contribution, of considerable practical value, and well suited to publication in PLoS Computational Biology.

Overall, the paper is well written, and makes clear the connection with the "phase type" method, which has been developed independently. However, the explanations could be yet clearer. In particular, the captions to the figures are too brief: it would be really helpful to walk the reader through the figures in the main text. In particular, the text above 2.1.1 needs to be clarified and expanded.

Specific comments:

- Note that the GF method also can include multiple recombining loci, though there is still the same computational difficulty as with two-way migration.

- Bidirectional process can be handled using a series expansion - which may be worth noting

119 -Define the term CAS

Eq 2 - Was the inverse Laplace Transform defined?

Fig 2 caption should refer to Fig 1 not Fig 2.2; similarly in line 150 there is a refrence to Fig 2.2C - should be Fig 2C

166 - The explanation of the bSFS is hard to follow.

236 What is a 3Ne model??

240 - specify "one-way migration"

253 - delete "a"

289 "But can be" - word missing?

299 - There is a statement here that suggests that phase-type theory avoids computational difficulties with bi-directional migration. However, I suspect that it requires matrix inversion, which is expensive.

**Have the authors made all data and (if applicable) computational code underlying the findings in their manuscript fully available?**

Reviewer #1: Yes

Reviewer #2: Yes

Reviewer #3: Yes

PLOS authors have the option to publish the peer review history of their article (what does this mean?). If published, this will include your full peer review and any attached files.

Reviewer #1: **Yes: **Jeffrey P. Spence

Reviewer #2: No

Reviewer #3: **Yes: **Nick Barton

Figure Files:

Data Requirements:

Reproducibility:

References:

---

## [Decision Letter · Decision Letter 1]

1 Sep 2022

Dear Mr Bisschop,

We are pleased to inform you that your manuscript 'Graph-based algorithms for Laplace transformed coalescence time distributions.' has been provisionally accepted for publication in PLOS Computational Biology.

Best regards,

Mark M. Tanaka

Academic Editor

PLOS Computational Biology

William Noble

Section Editor

PLOS Computational Biology

Reviewer 2 has some remaining issues to for you to consider if you wish.

Reviewer's Responses to Questions

**Comments to the Authors:**

Reviewer #1: The author has satisfactorily addressed all of my previous concerns.

Reviewer #2: The author has addressed nearly all of my comments. The mathematical descriptions are now clearer, and the notation is carefully spelled out and internally consistent. There is a convincing experiment to verify the accuracy of output.

The software will be useful and offer some noticeable gains in efficiency in the generating function approach to computing sample configuration probabilities. There are still some obvious limitations, such as in scalability with sample size and the current lack of the possibility of cycles in the underlying graph, but these are clearly flagged and discussed.

I would still like to see an enquiry into a question that is biologically driven. The author notes their intention to report a real world application elsewhere, but there is room for something here without having to go into all the details about data pre-processing. This paper is still mainly focused on algorithms, data structures, and computational issues. These are all important and relevant, but for papers appearing in a journal such as PLOS Computational Biology my feeling is that it should go further than this. What have we learned about, say, historical migration events that we didn't know before? (Okay, I'll leave it there. I can see that the highly esteemed other reviewers do not seem so exercised about this, and I won't make it a hill to die on.)

Minor comments:

l6 - duplicated 'random random'

l129-130 - is this usual notation for dot product? I would expect to see p.r or <p,r>.

Fig 3 caption - 7e-7 formatting looks a bit misleading. Better to write something in full, e.g. $7 \\times 10^{-7}$.</p,r>

Reviewer #3: This revision deals well with my suggestions for clarification. Also, I think that Bisschop's argument for publishing this as a separate paper, rather than bundling it with the gimble paper, is convincing.

**Have the authors made all data and (if applicable) computational code underlying the findings in their manuscript fully available?**

Reviewer #1: Yes

Reviewer #2: Yes

Reviewer #3: Yes

PLOS authors have the option to publish the peer review history of their article (what does this mean?). If published, this will include your full peer review and any attached files.

Reviewer #1: **Yes: **Jeffrey P. Spence

Reviewer #2: No

Reviewer #3: No

---

## [Editor Report · Acceptance letter]

10 Sep 2022

PCOMPBIOL-D-22-00768R1 

Graph-based algorithms for Laplace transformed coalescence time distributions.

Dear Dr Bisschop,

I am pleased to inform you that your manuscript has been formally accepted for publication in PLOS Computational Biology. Your manuscript is now with our production department and you will be notified of the publication date in due course.

With kind regards,

Zsofia Freund
